# HIV-1 controllers possess a unique CD8+ T cell activation phenotype and loss of control is associated with increased expression of exhaustion markers

Amber D. Jones[1,2‡], Zachary Capriotti[2,3‡], Erin Santos[4], Angel Lin[2], Rachel Van Duyne[2,5], Stephen Smith[4], Zachary Klase[2,5,6*]

1 Department of Biological Sciences, University of the Sciences, Philadelphia, Pennsylvania, United States of America, 2 Department of Pharmacology and Physiology, Drexel University College of Medicine, Philadelphia, Pennsylvania, United States of America, 3 Molecular and Cell Biology and Genetics Graduate Program, Drexel University College of Medicine, Philadelphia, Pennsylvania, United States of America, 4 The Smith Center for Infectious Diseases and Urban Health, West Orange New Jersey, United States of America, 5 Center for Neuroimmunology and CNS Therapeutics, Institute for Molecular Medicine and Infectious Diseases, Drexel University College of Medicine, Philadelphia, Pennsylvania, United States of America, 6 Sidney Kimmel Cancer Center, Thomas Jefferson University, Philadelphia, Pennsylvania, United States of America

‡ These authors share first authorship on this work.
* zk76@drexel.edu

## Abstract

### Background

HIV-1 controllers are a rare population of individuals that exhibit spontaneous control of HIV-1 infection without antiretroviral therapy. Understanding the mechanisms by which HIV-1 controllers maintain and eventually lose this ability would be highly valuable in HIV-1 cure or vaccine research. Previous work revealed the ability of CD8+ T cells isolated from HIV-1 controllers to suppress HIV-1 replication in matched CD4+ T cells and PBMCs *ex vivo* and suggested the loss of control may be tied to CD8+ T cell exhaustion.

### Results

We explored whether CD8+ T cell exhaustion plays a role in the maintenance and loss of control by examining immune characteristics of HIV-1 persistent controllers and transient controllers who lost control within the duration of the study. Using flow cytometry, we analyzed exhaustion marker expression on CD8+ T cells from HIV-1 controllers and determined that they maintain a unique exhaustion profile as compared to people without HIV-1 and HIV-1 standard progressors. The low level of T cell exhaustion seen in HIV-1 controllers was reversed when these individuals lost control and showed increased viral loads. Combinatorial immune checkpoint blockade targeting exhaustion markers was able to restore *ex vivo* control in CD8+ T cells from former controllers.

**Data availability statement:** Values used to perform analyses and to generate graphs can be found in the Supporting Information files. Flow cytometry data files are available at the Harvard Dataverse Repository: https://dataverse.harvard.edu/dataset.xhtml?persistentId=doi:10.7910/DVN/TGVGZO.

**Funding:** Funding for this work came from Drexel University internal funds to ZK, and NIH grants NIH/NIDA R01-DA057337 and NIH/NIDA DP2-DA044550 to ZK from the National Institute on Drug Abuse - https://nida.nih.gov/ The funders played no role in the study design, data collection and analysis, decision to publish, or preparation of the manuscript.

**Competing interests:** The authors declare that they have no competing interests.

## Conclusions

These results suggest that CD8$^+$ T cell exhaustion compromises the ability to control viral replication in HIV-1 controllers. The character of exhaustion in response to HIV-1 and therapy is distinct in HIV-1 persistent controllers, transient controllers and standard progressors.

## Introduction

In the typical HIV-1 clinical progression without antiretroviral therapy (ART), HIV-1 standard progressors (SP ART-) experience a decline in immune function over time [1]. Within the first 6−8 weeks of infection, a significant decrease in CD4$^+$ T cells, a concomitant increase in viral load, and a corresponding increase in immune activation is observed [2–4]. To control the infection, a CD8$^+$ T cell-mediated adaptive immune response is mounted but is not sufficient to eradicate the virus [5,6]. Although ART effectively suppresses viral replication and can restore CD4$^+$ T cell counts, the virus persists and causes long-term chronic inflammation and immune activation [7–12]. While most SP ART- lose full immune function within 8−10 years post-infection, a rare group of individuals deemed "HIV-1 controllers" spontaneously suppress viral replication without ART, yielding significantly slower disease progression, maintenance of normal CD4$^+$ T cell counts, and improved life expectancy [2,13–17]. Despite this remarkable ability, a subset of HIV-1 controllers deemed "transient controllers" (TC) eventually lose their ability to suppress viral replication, typically prompting therapeutic intervention with ART [18]. Since the identification of HIV-1 controllers, there has been strong interest in determining the underlying host and virological mechanisms behind this phenotype, which could inform the development of a functional cure or vaccine that mimics this natural effect. Understanding the loss of control in TC specifically could potentially help elucidate these underlying mechanisms.

Varying regulation of T cell activation and exhaustion are likely at play in the maintenance and loss of control in HIV-1 controllers [19,20]. During acute infection, activated T cells respond with cytokine secretion, clonal expansion, degranulation, and upregulation of immune checkpoint receptors (ICRs) [21]. ICR signaling attenuates T cell activation as the infection is resolved, preventing tissue damage and maintaining homeostasis [22,23]. During chronic infections, T cell activation persists, and the immune system must adapt to minimize tissue damage while maintaining control of the infection, eventually leading to hypo-responsiveness and immunological tolerance of the virus [24,25]. T cell exhaustion is a mechanism by which this occurs, characterized by prolonged upregulation of ICRs resulting in hierarchical loss of T cell function [26–30]. Exhaustion directly impacts the ability of CD8$^+$ T cells to suppress viral replication, and these exhausted T cells can be identified by their unique expression and co-expression of multiple ICRs [22,31–37]. ICRs such as programmed cell death 1 (PD-1) and T cell immunoglobulin and mucin domain-3 (Tim-3) are classic CD8$^+$ T cell exhaustion markers implicated in chronic viral infections like HIV-1 and

certain cancers, and their co-expression is thought to represent the most severely exhausted phenotype [38–43]. Other ICRs implicated in T cell exhaustion include cytotoxic lymphocyte antigen-4 (CTLA-4), T cell immune receptor with Ig and ITIM domains (TIGIT), and lymphocyte-activation gene 3 (LAG3) [37]. TIGIT and LAG3 are known to be upregulated in CD8[+] T cells of people living with HIV-1 (PLWH) [44,45], while CTLA-4 seems to be upregulated only in CD4[+] T cells [46]. If and how these markers are involved in the maintenance and/or loss of viral control in HIV-1 controllers is still poorly understood.

Our previous work identified that the ability of HIV-1 controllers to suppress viral replication *in vivo* was accompanied by the ability to suppress HIV-1 replication *ex vivo* in total PBMC cultures as well as CD4[+]/CD8[+] T cell co-cultures [19]. This finding confirmed similar reports shown in CD4[+]/CD8[+] T cell co-culture models [20,28]. We then showed that the ability to suppress viral replication *ex vivo* was mediated by CD8[+] T cells, and that CD8[+] T cells isolated from blood samples pre-loss of control could suppress viral replication in autologous CD4[+] T cells isolated post-loss of control in TC who otherwise lost the ability to suppress viral replication *ex vivo* [19]. In a limited panel of activation and exhaustion markers, we observed low levels of activation markers CD38 and HLA-DR and exhaustion marker PD-1 on CD8[+] T cells from HIV-1 controllers and these levels were indistinguishable from those of people without HIV-1 (PWoH) [19]. Interestingly, we observed a significant increase in PD-1 expression after the loss of control in TC, suggesting the maintenance of low CD8[+] T cell exhaustion is important for maintaining the ability to control viral replication in HIV-1 controllers [19].

During this study, several HIV-1 controllers monitored in our cohort lost the ability to suppress viral replication *in vivo* and were subsequently treated with ART, revealing themselves as TC (Table 1). Longitudinal samples collected during periods of successful control, and after the loss of control and subsequent ART treatment gave us the opportunity to further explore the changes in CD8[+] T cell activation and exhaustion that occur when control is lost. In this study, we further explored T cell activation and exhaustion phenotypes in HIV-1 controllers with a wider panel of markers, revealing unique populations across participant groups. We also observed increased CD8[+] T cell exhaustion that was concurrent with the loss of control in TC and that these changes are not reversed over time on ART. Further, we demonstrated that combinatorial immune checkpoint blockade (cICB) targeting exhaustion markers PD-1 and Tim-3 can restore the ability to suppress viral replication *ex vivo* in TC after the loss of control.

## Materials and methods

### Sex as a biological variable

Our study examined both male and female study participants, and similar findings were reported for both sexes. However, the sample size presented does not specifically power our study to analyze differences between the sexes and these results may have sex-based differences in a larger sample.

### Study cohort

All study participants were recruited by the Smith Center for Urban Health and Infectious Disease. The Smith Center for Urban Health and Infectious Disease, East Orange, NJ obtained written informed consent for the collection of blood donations from study participants. Samples were collected by trained medical staff under approved University of the Sciences' protocol (IRB protocol 900702−3 and 797649−3). Recruitment began on 5/12/2015 and ended on 12/07/2017.

Two cohorts of people living with HIV (PLWH) were recruited, HIV-1 controllers and standard progressors (SP) (see clinical characteristics at enrollment – Table 1). HIV-1 controllers are defined as HIV-1 seropositive individuals with viral loads <2,000 copies/mL and a CD4[+] T cell count >500 cells/mm$^3$ [47–49]. For this study, classification as an HIV-1 controller required viremic control for a duration of at least 12 consecutive months in the absence of ART [47–49]. HIV-1 seropositive participants with a documented history of high viral load above 5,000 copies/mL were designated as SP. For the HIV-1 controller cohort, eight HIV-1 seropositive ART naïve participants were recruited who were able to suppress HIV-1 infection independent of known protective HLA alleles (Table 2). Four of these HIV-1 controllers lost control and

**Table 1. Clinical characteristics at study enrollment.**

| Classification | | Participant ID | Viral load (copies/mL) | CD4 count (cells/mm³) | CD8 count (cells/mm³) | cART regimen |
|---|---|---|---|---|---|---|
| **SP cART+** standard progressors on cART | | ADV7140 | <20 | 497 | 593 | emtricitabine/tenofovir/dolutegravir |
| | | BRW1143 | <20 | 1,135 | 828 | emtricitabine/tenofovir/dolutegravir |
| | | XTD8730 | 30 | 291 | 764 | dolutegravir/atazanavir/ritonavir |
| | | KLU1328 | <20 | 269 | 584 | emtricitabine/tenofovir alafenamide/dolutegravir |
| | | UNJ7200 | <20 | 317 | 517 | emtricitabine/rilprivirine/tenofovir disoproxil fumarate |
| | | NLN8790 | <20 | 853 | – | efavirenz/emtricitabine/tenofovir disoproxil |
| | | QMY7270 | <20 | 1,367 | 1,292 | efavirenz/emtricitabine/tenofovir disoproxil |
| | | TLY1482 | <20 | 814 | 1,023 | elvitegravir/cobicistat/emtricitabine/tenofovir alafenamide |
| **SP cART-** standard progressors off cART | | KMJ1960 | 22,810 | 212 | 1,169 | n/a |
| | | WBZ1300 | 23,880 | 170 | – | n/a |
| | | ORL5590 | 2,880,820 | 62 | 344 | n/a |
| | | ARF1454 | 22,984 | 53 | 774 | n/a |
| | | UTF9050 | 11,390 | 995 | 419 | n/a |
| **VC** viremic controllers | **PC** persistent controllers | FJG8070 | 250 | 1,070 | 1,434 | n/a |
| | | HCQ6670 | 80 | 488 | 506 | n/a |
| | | SRS5930 | 230 | 699 | 792 | n/a |
| | | AEM9650 | 1,630 | 1,001 | 2,081 | n/a |
| | **TC** transient controllers | VQY4910 | 750/<20 | 723/ 1,025 | 987/ 735 | *elvitegravir/cobicistat/emtricitabine/tenofovir alafenamide |
| | | MPY1313 | 1,551/<20 | 473/ 571 | 743/ 954 | *emtricitabine/tenofovir disoproxil fumarate/dolutegravir |
| | | NIM1164 | 630/<20 | 769/ 896 | 1,950/ 1,726 | *abacavir/dolutegravir/lamivudine |
| | | FWU1270 | 160/<20 | 1,112/ 769 | 1,177/ 1,071 | *elvitegravir/cobicistat/emtricitabine/tenofovir alafenamide |
| **EC** elite controllers | | EXT1011 | <20 | 547 | 430 | abacavir/dolutegravir/lamivudine |
| **PWoH** people without HIV | | BTS1096 | – | 1,190 | 358 | n/a |
| | | CHT3368 | – | 360 | 646 | n/a |
| | | HFK1114 | – | – | – | n/a |
| | | ENS1452 | – | – | – | n/a |
| | | VIQ1422 | – | – | – | n/a |

"-"=data not available
"a/ b"=a: value at enrollment, b: value ~6 months after cART initiation (TC only)
"*"=participant was not on cART at study enrollment

began ART treatment during the course of this study and are thus deemed TC. Four HIV-1 controllers maintained control throughout the duration of the study and are thus deemed persistent controllers (PC). One HIV-1 controller is an elite controller (EC) who requested ART treatment, despite maintaining control, before the beginning of this study and thus

**Table 2. HLA allele genotyping.**

| Classification | Participant ID | HLA-A | HLA-A2 | HLA-B | HLA-B3 | HLA-C | HLA-C4 | DRB1 | DRB15 |
|---|---|---|---|---|---|---|---|---|---|
| **SP cART+** standard progressors on cART | ADV7140 | 03:01 | 33:03:00 | 07:TDVB | 58:01:00 | 07:02 | 07:18 | 09:CTZ | 15:ADZBV |
| | BRW1143 | 32:01:00 | 33:03:00 | 42:01:00 | 44:03:00 | 04:01 | 17:01 | 03:02 | 13:ASWXB |
| | XTD8730 | 33:03:01 | 66:01:01 | 15:16:01 | 44:03:01 | 04:01 | 14:02 | 01:02 | 15:03 |
| | KLU1328 | 01:01 | 03:01 | 07:TDVB | 38:01:00 | 07:02 | 12:03 | 13:AHUNK | – |
| | UNJ7200 | 03:01 | 31:01:00 | 07:TDVB | 53:01:00 | 04:01 | 07:18 | 07:FKP | – |
| | NLN8790 | 30:02:00 | 68:ASXJM | 53:01:00 | 57:03:00 | 04:01 | 0.75 | 03:02 | 15:03 |
| | QMY7270 | 02:02 | 30:01:00 | 13:02 | 35:ABRHF | 06:02 | – | 07:JDKZ | 13:ASVAC |
| | TLY1482 | – | – | – | – | – | – | – | – |
| **SP cART-** standard progressors off cART | KMJ1960 | – | – | – | – | – | – | – | – |
| | WBZ1300 | – | – | – | – | – | – | – | – |
| | ORL5590 | – | – | – | – | – | – | – | – |
| | ARF1454 | – | – | – | – | – | – | – | – |
| | UTF9050 | – | – | – | – | – | – | – | – |
| **PC** persistent controllers | FJG8070 | 23:CJT | 34:02:00 | 35:01:00 | 58:01:00 | 04:01 | 06:02 | 09:CTZ | 13:ASVAC |
| | HCQ6670 | 02:ANGA | 30:01:00 | 42:01:01 | 57:03:01 | 17:01 | 18:02 | 08:AFPMU | 13:AHUNK |
| | SRS5930 | 23:CJT | 29:02:00 | 45:01:01 | 81:01:00 | 05:01 | 08:04 | 12:DUKV | 13:ASVAC |
| | AEM9650 | 2:AMAUU | 2:AWUEM | 42:01:01 | 49:01:00 | 05:01 | 17:01 | 07:JDKZ | 11:02 |
| **TC** transient controllers | VQY4910 | 02:ANGA | 74:01:00 | 35:01:00 | 57:03:00 | 04:01 | 07:01 | 01:AETTD | 15:03 |
| | MPY1313 | *29 | *31 | *18 | *39 | *12 | *12 | *11 | *15 |
| | NIM1164 | 24:02:00 | 30:01:00 | 14:02:01 | 42:01:01 | 08:02 | 17:01 | 01:02 | 03:02 |
| | FWU1270 | 02:ANGA | 11:01 | 18:01 | 35:01:00 | 07:04:01 | 16:01:01 | 03:02 | 12:02 |
| **EC** elite controllers | EXT1011 | 23:CJT | 74:01:00 | 52:01:00 | 58:01:00 | 03:02 | 16:01 | 13:04 | 13:ASVAC |
| **PWoH** people without HIV | BTS1096 | 29:02:00 | 33:01:00 | 14:02 | 44:03:00 | 08:02 | 16:01 | 01:02 | 07:JDKZ |
| | CHT3368 | 30:02:00 | 31:16:00 | 35:03:00 | 57:03:00 | 12:03 | 18:02 | 13:APENP | 16:AFNDC |
| | HFK1114 | 30:02:00 | 34:02:00 | 15:03:01 | 44:03:01 | 02:10 | 04:01 | 04:05 | 15:03 |
| | ENS1452 | 02:ANGA | 68:02:00 | 44:ANAW | 45:01:00 | 05:01 | 16:01 | 03:ADEKP | 15:ADZBV |
| | VIQ1422 | 23:CJT | 24:AUJRX | 07:TDVB | 35:03:00 | 07:02 | 0.50 | 03:AWXCS | 14:AHVBM |
| | | *=results were considered "low resolution" | | | | | | | |
| multiple allele codes: | | "CJT"=01/17, "ANGA"=02:01/02:01L, "AMAUU"=02/603, "ATRDH"=01/665, "ASXJM"=02/163, "AWUEM"=01/102/665, "AUJRX"=02/353, "TDVB"=07:02/07:61/07:161N, "ANAW"=44:02/44:02S, "ABRHF"=01/332, "CTZ"=01/21, "AHUNK"=01/117/166/173/190, "FKP"=01/34, "JDKZ"=01/34/72, "AFPMU"=04/59, "DUKV"=01/06/10/17, "AETTD"=01/50/67, "APENP"=02/208/212, "ADEKP"=03:01/03:68N/03:83/03:104, "AWXCS"=03:01/03:42/03:68N/03:83/03:87/03:104/03:124/03:127/03:132/03:135/03:137, "ADZBV"=15:01/15:86/15:110/15:113N, "ASWXB"=02/67/208/212, "ASVAC"=01/117/166/173/190/215/218, "AFNDC"=02/22/35/39, "AHVBM"=54/113/125/157 | | | | | | | |

was on ART treatment throughout the duration of this study. The SP cohort included eight HIV-1 seropositive participants on ART (SP ART+) and five ART naïve HIV-1 seropositive participants (SP ART-). Five age-matched HIV-1 seronegative participants were recruited (PWoH). Both PC and TC groups are considered viremic controllers (VC) as they displayed detectable viremia over the course of the study while the EC maintained viremia below the detectable limit. Susceptibility to infection and classification of *in vivo* and *ex vivo* control in this cohort has already been published [19].

## PBMC isolation and activation

PBMCs were isolated from whole blood samples using Ficoll (GE Healthcare) gradient centrifugation. Briefly, whole blood was spun at 1500 rpm for 5 min at 21°C and plasma was removed and stored in 1 mL aliquots in liquid nitrogen. Blood

was then diluted 1:1 in phosphate buffered saline (PBS) without $Ca^{2+}$ and $Mg^{2+}$ (GenClone) and 15 mL of diluted blood was carefully overlaid onto 13 mL Ficoll and spun at 1500 rpm for 30 min at 21° C without braking. PBMCs were collected from the interphase and washed 2X with 10 mL PBS without $Ca^{2+}/Mg^{2+}$. PBMCs were counted and aliquoted in 90% heat inactivated fetal bovine serum (HI-FBS) (HyClone) containing 10% dimethyl sulfoxide (DMSO) (Fisher) for cryopreservation in liquid nitrogen. Frozen PBMCs were thawed rapidly and activated in RPMI-1640 complete media (GenClone) supplemented with 20% HI-FBS (HyClone), 1% penicillin-streptomycin-glutamine (PSQ) (ThermoFisher Scientific), 5% (5 U/mL) human rIL-2 (NIH AIDS Reagent Program, cat# 136), and 5 μg/mL phytohemagglutinin-P (PHA-P) (Sigma, cat# L1668) for 48 h at 37°C and 5% $CO_2$. After activation, PHA-P-containing media was removed, cells were counted, viability was determined by trypan blue stain, and cells were used for downstream experiments.

## CD4+ and CD8+ T Cell isolation and activation

PBMCs prepared as described above were used for T cell isolation. Briefly, PBMCs were thawed and CD4+ and CD8+ T cells were isolated using Miltenyi MACS negative isolation kits (cat# 130-096-533 and cat# 130-096-495, respectively) according to the manufacturer's protocol. Purity of isolated cells was assayed by flow cytometry as indicated below. Cells are routinely isolated at >80% purity. Negatively selected CD4+ and CD8+ T cell populations were independently activated in RPMI-1640 complete media (GenClone) supplemented with 20% HI-FBS (HyClone), 1% PSQ (ThermoFisher Scientific), 5% (5 U/mL) human rIL-2 (NIH AIDS Reagent Program, cat# 136), and 5 μg/mL PHA-P (Sigma, cat# L1668) for 48 h at 37°C and 5% $CO_2$. After activation, PHA-P-containing media was removed, CD4+ and CD8+ T cell cultures were counted, viability was determined by trypan blue stain, and cells were used for downstream experiments.

## HIV-1 virus stock

The HIV-1 stock used in this study was generated by transfecting pNL4−3 (NIH AIDS Reagent Program, ARP-2852, contributed by Dr. M. Martin) into HEK293T cells [American Type Culture Collection (ATCC, CRL-11268)] using Trans-Fectin Lipid Reagent (BioRad, Cat# 1703351) following manufacturer's instructions. Transfected cells were cultured in Dulbecco's Modified Eagle's Medium (DMEM) supplemented with 10% HI-FBS (HyClone) and 1% PSQ (ThermoFisher Scientific) for 48 h at 37°C and 5% $CO_2$. After 48 h, virus containing supernatants were aspirated from the cells and spun at 1500 rpm for 5 min, supernatant was then aspirated and filtered through a 0.22 μm filter, and frozen in liquid nitrogen in 1 mL aliquots. Virus stocks were titered by infecting TZMbl cells (NIH AIDS Reagent Program, cat#8129) at two-fold serial dilutions and determining the HIV-1 p24 gag protein concentration by p24 ELISA (Zeptometrix, cat# 0801200) to estimate infectious units/mL. The virus stock used in this study was determined to contain $1.29 \times 10^5$ IU/mL and all infections were carried out at MOI of 0.01.

## Immune checkpoint blockade viral suppression assay

Isolated activated CD4+ T cells were prepared as described above and cultured alone as a control. Isolated activated CD4+ and CD8+ T cells were prepared as described above and co-cultured at a 1:1 ratio for 3−6 h at 37°C and 5% CO2 before subsequent treatments. Activated total PBMCs (prepared as described above) or isolated activated CD4+ and CD8+ T cells were either left untreated or treated with anti-PD-1 (BioLegend, Ultra LEAF purified human anti-PD-1 antibody, clone EH12.2H7) and/or anti-Tim-3 (CD366) antibody (BioLegend, UltraLEAF purified anti-human, clone F38-2E2) immune checkpoint blockade for 24 h at 37°C and 5% $CO_2$. 24 h after ICB or mock treatment, cells were infected with HIV-1 NL4−3 virus (prepared as described above) at MOI = 0.01 for 24 h at 37°C and 5% $CO_2$. After 24 h, the virus containing media was removed and replaced with fresh ICB-containing media. Supernatants were collected and fresh ICB-containing media was replenished every 48 h post infection for 6 days. Virus production was evaluated by measuring the p24 levels in supernatant using ELISA (Zeptometrix, cat# 0801200) according to manufacturer's instructions and % suppression of viral replication was calculated as 100 – ((p24 concentration of ICB treated/ p24 concentration of ICB

untreated) x 100). Media used post-activation was RPMI-1640 complete media (GenClone) supplemented with 20% HI-FBS (HyClone), 1% PSQ (ThermoFisher Scientific), 5% (5 U/mL) human recombinant IL-2 (NIH AIDS Reagent Program, cat# 136) and supplemented with various treatments as specified above.

## Flow cytometry

Cells were washed with PBS without $Ca^{2+}/Mg^{2+}$ then incubated in 2 mL DNAse (NEB, cat# m030s) for 10 min on ice. Cells were then washed with PBS and stained for viability using Zombie Yellow live/dead stain (Biolegend, cat# 423103) at a 1:1000 dilution in PBS without $Ca^{2+}/Mg^{2+}$ for 10 min on ice protected from light. Cells were washed again and incubated for 30 min on ice protected from light with various combinations of the following antibodies: CD3 (BD Biosciences (BD), Alexa700, clone SP34−2), CD4 (BD, PerCP-Cy5.5), CD8 (BD, BV786, clone RPA-T8), CD279 (PD-1) (Invitrogen, PE-Cyanine7, clone eBioJ105), human Tim-3 (R&D Systems, Alexa Fluor 488, clone 344823), CD223 (LAG3) (Invitrogen, eFluor 450, clone 3DS223H), TIGIT (Invitrogen, PerCP-eFluor710, clone MBSA43), CD152 (CTLA-4) (Invitrogen, APC, clone 14D3), HLA-DR (BD, APC-H7, clone L243), and CD38 (BD, PE, clone HIT2). Cells were then washed 2x with PBS without $Ca^{2+}/Mg^{2+}$ and incubated with 1% formaldehyde in PBS without $Ca^{2+}/Mg^{2+}$ for 20 min at room temperature protected from light. Cells were washed once more with PBS without $Ca^{2+}/Mg^{2+}$ and then run on a Cytek FACSort DxP12 flow cytometer. Data was analyzed using FlowJo v10.6.1 software. Data was calculated as the percent of live (Zombie yellow⁻), CD3⁺ singlets positive for the given marker of interest.

## Intracellular cytokine staining

Frozen PBMCs were thawed in RPMI supplemented with 20% HI-FBS (HyClone) and 1% PSQ (ThermoFisher Scientific), plated at $1 \times 10^6$ cells per condition, and allowed to rest overnight at 37°C and 5% $CO_2$. Cells were unstimulated (DMSO control), stimulated with PMA/Ionomycin (50ng/mL, Sigma #P1585; 500ng/mL, Sigma #10634, respectively), or Gag peptide pool (2 µg/mL/peptide, BEI Resources, NIAID, NIH: Peptide Pool, Human Immunodeficiency Virus Type 1 Subtype B (Consensus) Gag Region, HRP-12425) for 1 h at 37°C and 5% $CO_2$. All cell conditions received co-stimulatory antibody treatment – anti-CD28 (1 µg/mL; BD #556620) and anti-CD49d (1 µg/mL; BD #555502), regardless of stimulation. Cells were treated with GolgiPlug (1µl/condition; BD #555029) to prevent extracellular secretion of cytokines and were incubated for 4.5 h at 37°C and 5% $CO_2$. Cells were then prepared for flow cytometry as indicated above for viability (Zombie Yellow) and extracellular staining (CD3, CD4, CD8) followed by fixation and permeabilization according to manufacturer's instructions (BD #554714). Cells were maintained in Perm/Wash Buffer and stained for intracellular cytokines for 30 min on ice protected from light with the following antibodies: TNF-α (BD, PE, clone MAb11 #557068), IL-2 (BD, FITC, clone MQ1-17H12 #559361), and IFN-ɣ (BD, Alexa Fluor 647, clone 4S.B3 #563495). Cells were then washed 2x with Perm/Wash buffer and incubated with 1% formaldehyde in PBS without $Ca^{2+}/Mg^{2+}$ for 20 min at room temperature protected from light. Cells were washed once more with PBS and then run on a Cytek FACSort DxP12 flow cytometer. Data was analyzed using FlowJo v10.6.1 software. Data was calculated as the percent of live (Zombie yellow⁻), CD3⁺ singlets positive for the given marker of interest: IL-2⁺, TNF-α⁺ (IFN-ɣ⁺TNF-α⁺ + IFN-ɣ⁻TNF-α⁺), IFN-ɣ⁺ (IFN-ɣ⁺TNF-α⁺ + IFN-ɣ⁺TNF-α⁻), TNF-α ⁺IFN-ɣ ⁻, TNF-α ⁻IFN-ɣ ⁺, TNF-α ⁺IFN-ɣ ⁺.

## Multivariate clustering analysis

Clustering and t-Distributed Stochastic Neighbor Embedding (tSNE) dimensionality reduction of flow cytometry data was performed using the FlowSOM and tSNE functions available within FlowJo v10. In brief, the category variable was used to assign an identifier to each individual participant sample to allow examination of specific groups after concatenation and analysis. Cells from each participant sample that were identified as CD3⁺Zombie Yellow⁻ singlets were combined into a single data file using the concatenate function with a cutoff of 20,000 events per individual sample. tSNE was performed on this combined data set focusing on Tim-3, CD38, TIGIT, PD-1, CTLA-4, HLA-DR, LAG3, CD8 as variables of interest.

Clustering was performed for an assumed 10 clusters (number of stained variables plus two) using FlowSOM and a heatmap was generated. Identified clusters were then mapped back onto the tSNE plot. Selection of the individual participants using the category variable as assigned before concatenation was used to visualize groupings by participant group. To examine the expression of individual markers, intensity of staining was mapped onto the tSNE plot as a third dimension.

## Statistics

Data analysis for graphs presented in Figs 1–3, 6, 7 and S1–S3 was performed in GraphPad Prism v10.2.1. Data for Figs 1–3, 6, 7E–7I and S1–S3 are presented as graphs of all individual data points with a line representing the mean and p-value thresholds as noted for each figure while Fig 7B–7D are graphed with lines only. Specific statistical tests performed for each figure are noted in the figure legends and p-values are noted in the text.

## Results

### HIV-1 persistent controllers display unique CD8+ T cell exhaustion profiles

We hypothesized that the ability of HIV-1 controllers to suppress HIV-1 replication in the absence of ART is related to their relatively low levels of CD8+ T cell exhaustion [19,50]. To examine this, we performed flow cytometry on PBMCs from PC and TC during periods of successful control (herein referred to as VC for simplicity) to characterize their exhaustion and activation phenotypes compared to PBMCs from SP ART- and PWoH (Table 1). Interestingly, both VC and PWoH displayed a similar activation profile (Fig 1A) characterized by significantly lower percentages of CD38+ (p = 0.009, p = 0.003

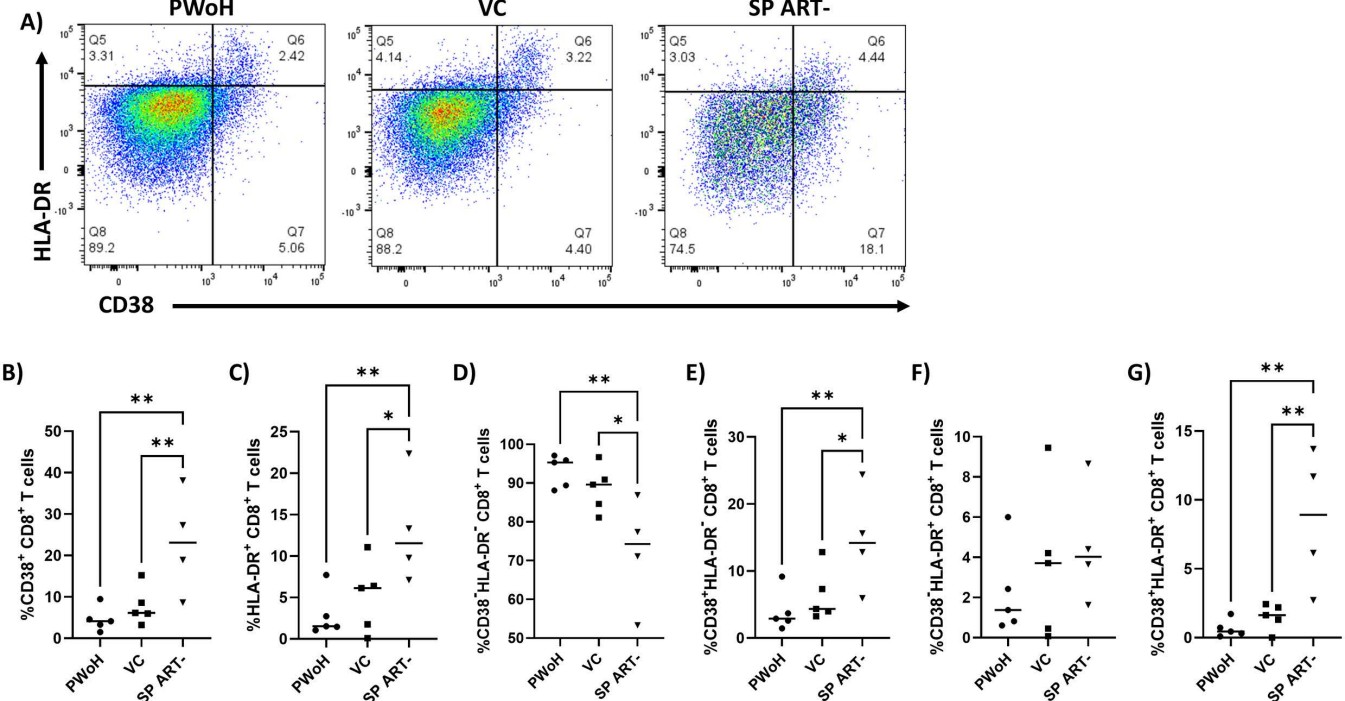

**Fig 1. HIV-1 persistent controllers display low levels of CD8+ T cell activation markers.** PBMCs were isolated from blood samples of PWoH (n = 5), VC (n = 5) and SP ART- (n = 4) and flow cytometry was performed gating for live (Zombie Yellow⁻), CD8+ T cells (CD3+CD8+). **A)** Representative gating of CD8+ T cells analyzed for the expression of CD38 (x axis) and HLA-DR (y axis). Percentage of live CD8+ T cells that are **B)** CD38+, **C)** HLA-DR+, **D)** CD38⁻HLA-DR⁻, **E)** CD38+HLA-DR⁻, **F)** CD38⁻HLA-DR+, **G)** CD38+HLA-DR+. Ordinary one-way ANOVA with uncorrected Fisher's LSD was performed to determine statistical significance. * p ≤ 0.05, ** p ≤ 0.01.

respectively) (Fig 1B), HLA-DR⁺ (p=0.03, p=0.007 respectively) (Fig 1C), CD38⁺HLA-DR⁻ (p=0.03, p=0.008 respectively) (Fig 1E), and CD38⁺HLA-DR⁺ (p=0.003, p=0.001 respectively) (Fig 1G) CD8⁺ T cells compared to SP ART- while CD38⁻ HLA-DR⁺ CD8⁺ T cell levels were not significantly different across groups (p=0.5, p=0.3, p=0.6) (Fig 1F). Conversely, both VC and PWoH maintained significantly higher percentages of CD38⁻HLA-DR⁻ CD8⁺ T cells than SP ART- (p=0.02, p=0.004 respectively) (Fig 1D). The high level of activation markers expressed on CD8⁺ T cells in SP ART- confirms previous reports that chronic HIV-1 infection is associated with increased immune activation [51–57]. This data also confirms a previous report that HIV-1 VC, despite exhibiting low-level detectable viremia (Table 1), maintain immune activation states comparable to PWoH [58].

We next examined the expression of exhaustion markers PD-1 and Tim-3 on CD8⁺ T cells from PWoH, VC, and SP ART- (Fig 2A). PD-1 does not restrict cytotoxic activity in the acute phase of infection, but during chronic infection, increased PD-1 expression on CD8⁺ T cells results in exhaustion and failure to suppress HIV-1 replication [39,59–62]. Tim-3 has also been shown to be upregulated in chronic HIV-1 infection and its expression correlates to clinical parameters of disease progression [40]. Like the activation marker data (Fig 1), VC and PWoH displayed similar exhaustion phenotypes characterized by significantly lower percentages of PD-1⁺ (p=0.005, p=0.008 respectively) (Fig 2B), PD-1⁺Tim-3⁻ (p=0.001, p=0.002 respectively) (Fig 2F), and PD-1⁺Tim-3⁺ (p=0.005, p=0.02 respectively) (Fig 2G) CD8⁺ T cells compared to SP ART- and conversely, significantly higher percentages of PD-1⁻Tim-3⁻ CD8⁺ T cells (p=0.003, p=0.004 respectively) (Fig 2D). Levels of Tim-3⁺ CD8⁺ T cells were not significantly different across groups, but SP ART- appears to trend higher than VC and PWoH (p=0.06, p=0.1 respectively) (Fig 2C). VC also displayed significantly lower

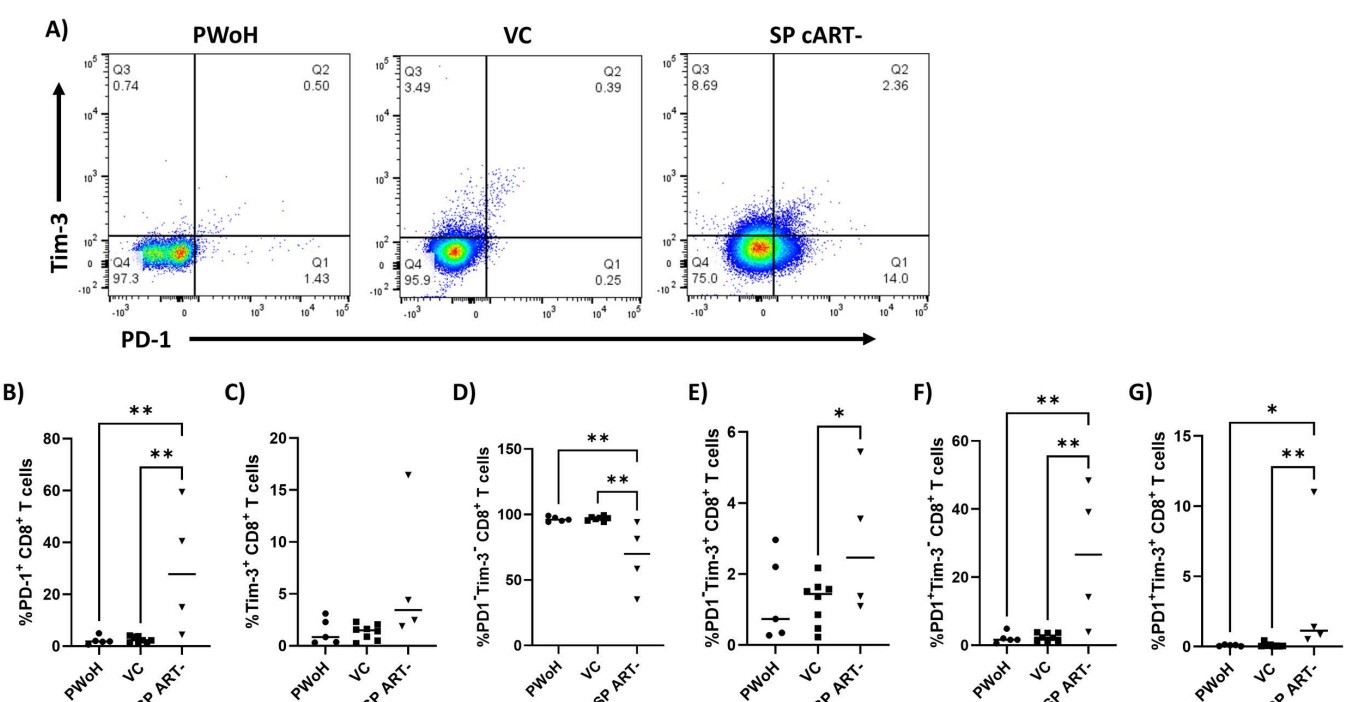

**Fig 2. HIV-1 persistent controllers display low levels of CD8⁺ T cell exhaustion markers PD-1 and Tim-3.** PBMCs were isolated from blood samples of PWoH (n=5), VC (n=8) and SP ART- (n=4) and flow cytometry was performed gating for live (Zombie Yellow⁻), CD8⁺ T cells (CD3⁺CD8⁺). **A)** Representative gating of CD8⁺ T cells analyzed for the expression of PD-1 (x axis) and Tim-3 (y axis). Percentage of live CD8⁺ T cells that are **B)** PD-1⁺, **C)** Tim-3⁺, **D)** PD-1⁻Tim-3⁻, **E)** PD-1⁻Tim-3⁺, **F)** PD-1⁺Tim-3⁻, and **G)** PD-1⁺Tim-3⁺. Ordinary one-way ANOVA with uncorrected Fisher's LSD was performed to determine statistical significance. * p≤0.05, ** p≤0.01.

levels of PD-1⁻Tim-3⁺ CD8⁺ T cells compared to SP ART-, while a similar but non-significant trend exists for PWoH versus SP ART- (p = 0.05, p = 0.08 respectively) (Fig 2E). As in Fig 1, no significant differences existed between VC and PWoH for any comparison (p => 0.9, p => 0.9, p => 0.9, p = 0.9, p => 0.9, p = 0.8 respectively) (Fig 2B–2G), further suggesting VC maintain activation and exhaustion states comparable to PWoH during periods of successful control.

To further explore CD8⁺ T cell exhaustion phenotypes in VC, we examined additional exhaustion markers CTLA-4, LAG3, and TIGIT in a limited number of participants and compared their expression in VC to PWoH and a selection of standard progressors suppressed on therapy (SP ART+) (S1 Fig). Breaking the pattern observed in Figs 1 and 2, VC displayed an altered exhaustion profile compared to PWoH and SP ART+ characterized by significantly higher percentage of CTLA-4⁺ (p = 0.01, p = <0.0001 respectively) (S1A Fig) and LAG3⁺ (p = 0.001, p = <0.0001 respectively) (S1B Fig) CD8⁺ T cells. The observed CTLA-4 expression pattern may be explained by the fact that the HIV-1 Nef protein, which may be highly expressed in SP ART+ and less expressed in VC, has been shown to downregulate CTLA-4 [63]. Additionally, VC displayed significantly lower percentage of TIGIT⁺ CD8⁺ T cells compared to PWoH (p = 0.002), but no significant difference was observed compared to SP ART- (p = 0.1) (S1C Fig). Taken together, these data support the conclusion that VC display a unique exhaustion phenotype compared to SP ART- and PWoH. Specifically, they maintain low levels of Tim-3⁺, PD-1⁺, and TIGIT⁺ CD8⁺ T cells, while showing increased enrichment of CTLA-4⁺ and LAG3⁺ CD8⁺ T cells.

### Loss of control is concurrent with increased CD8⁺ T cell exhaustion

Our previous study demonstrated that PBMCs from HIV-1 controllers could suppress HIV-1 viral replication *ex vivo* due to the activity of CD8⁺ T cells and that this ability was lost in three TC after losing their ability to control [19]. Over the course of this study, several TC lost their ability to control and were placed on ART. To better understand whether CD8⁺ T cell activation and/or exhaustion is associated with this loss of control, we examined CD38, HLA-DR, PD-1, and Tim-3 expression on CD8⁺ T cells from TC post loss of control (TC – post control) compared to VC during periods of successful control (VC – control). Interestingly, no significant differences were observed in the levels of CD8⁺ T cell activation marker expression upon loss of control (p = 0.6, p = 0.7, p = 0.5, p = 0.5, p = 0.6, p => 0.9 respectively) (Fig 3A–3F). However, CD8⁺ T cell exhaustion levels were markedly increased in TC – post control as seen by significantly increased PD-1⁺ (p = 0.0003) (Fig 3G), Tim-3⁺ (p = <0.0001) (Fig 3H), PD-1⁺Tim-3⁻ (p = 0.0004) (Fig 3J), PD-1⁻Tim-3⁺ (p = <0.0001) (Fig 3K), and PD-1⁺Tim-3⁺ (p = <0.0001) (Fig 3L) CD8⁺ T cells and conversely, significantly lower levels of PD-1⁻Tim-3⁻ CD8⁺ T cells (p = <0.0001) (Fig 3I) compared to VC – control. Whether CD8⁺ T cell exhaustion was the cause, or a result of the loss of control is unclear. However, the observed concurrence of CD8⁺ T cell exhaustion and loss of control supports our hypothesis that the ability of HIV-1 controllers to suppress HIV-1 replication is driven by effective regulation of CD8⁺ T cell exhaustion.

### Multivariate analysis reveals heterogenous populations of T cells expressing various combinations of activation and exhaustion markers

To evaluate the importance of individual activation and exhaustion markers in a qualitative manner independent of gating choices, we performed clustering analysis using flow cytometry data from PWoH, PC and TC during successful control (VC – control), TC post control (TC – post control), SP ART-, and SP ART+ (Fig 4). Populations identified as live, CD3⁺ T cells were clustered using the FlowSOM plugin for FlowJo v10. This clustering revealed several populations of T cells displaying unique combinations of activation and exhaustion markers that were color coded and designated 1–10 (Fig 4A). From this data, a heatmap was generated depicting the contribution of each unique cluster to the overall makeup of the various participant groups (calculated as the percentage of cells from each participant group falling into a given cluster) (Fig 4B). From this analysis, we can see that during periods of successful control, VC – control resemble PWoH with similar abundance of clusters 7, 8, and 10, but upon loss of control, TC – post control more closely resemble SP ART+ with similar abundance of clusters 1 and 4 (Fig 4B). To better visualize these clusters and determine whether any are overrepresented in a particular group,

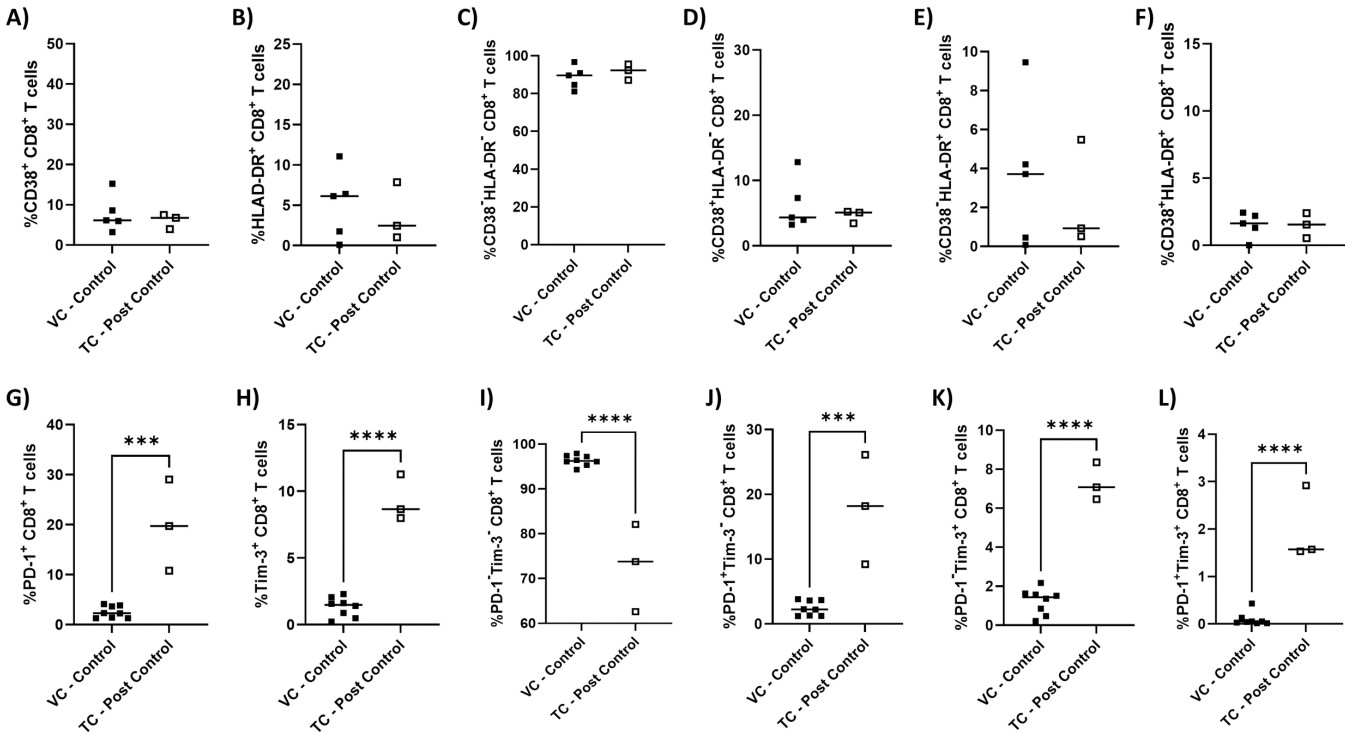

**Fig 3. Exhaustion, but not activation markers are increased after HIV-1 transient controllers lose control.** PBMCs were isolated from blood samples of VC – control (n = 5−8) or TC – post control (n = 3) and flow cytometry was performed gating for live (Zombie Yellow-), CD8⁺ T cells (CD3⁺CD8⁺). Graphed are the percentages of live CD8⁺ T cells that are **A)** CD38⁺, **B)** HLA-DR⁺, **C)** CD38⁻HLA-DR⁻, **D)** CD38⁺HLA-DR⁻, **E)** CD38⁻HLA-DR⁺, **F)** CD38⁺HLA-DR⁺, **G)** PD-1⁺, **H)** Tim-3⁺, **I)** PD-1⁻Tim-3⁻, **J)** PD-1⁺Tim-3⁻, **K)** PD-1⁻Tim-3⁺, and **L)** PD-1⁺Tim-3⁺. Unpaired two-tailed T tests were performed for each panel to determine statistical significance. *** p ≤ 0.0005, **** p ≤ 0.0001.

we performed dimensional reduction using t-Distributed Stochastic Neighbor Embedding (tSNE) (Fig 4C). The tSNE plot revealed that several of the clusters identified might be further broken down into sub clusters (i.e., cluster 7 or 8 which both occupy multiple distinct regions on the tSNE plot). By displaying cells from each participant group alone, we were able to determine which clusters and regions were shared or unique across groups (Fig 4D–4H). PWoH were characterized by the presence of CD8⁺ T cells (cluster 7) or CD4⁺ T cells (CD3⁺CD8⁻ cells, cluster 10) with low levels of CD38 and HLA-DR (Fig 4D). SP ART- displayed multiple populations of HLA-DR⁺ and CD38⁺ CD8⁺ T cells (cluster 2 and cluster 5) along with some smaller populations with low expression of HLA-DR and CD38 (cluster 7) (Fig 4E). VC – control displayed a profile with some similarities to both PWoH and SP ART-, but more closely resembled PWoH (Fig 4F). Interestingly, VC – control also possessed unique populations not seen in PWoH or SP ART- (labeled 1* and 8* as sub-populations of clusters 1 and 8 respectively, indicated by outline in Fig 2F). As the TC – post control were placed on ART, we also analyzed SP ART+ to account for any contribution of ART. Interestingly, TC – post control displayed profiles similar to SP ART +, with a unique region shared between them (labeled 4* indicated by outline in Fig 2G and 2H) representing CD8⁻ T cells with high expression of activation markers. TC – post control also displayed a unique population not seen in any other group (labeled 5* in Fig 2G). Taken together, these data show that each group contains distinct T cell populations with varying activation and exhaustion levels that could not be parsed out through traditional flow cytometry gating analysis alone.

To further characterize the phenotypes of T cells in regions unique to HIV-1 controllers, we displayed the expression of CD8, CD38, HLA-DR, PD-1, Tim-3, TIGIT, CTLA-4 and LAG3 as heatmaps projected onto the tSNE plot generated in Fig 4C (Fig 5). This showed that population 1*, which is unique to VC – control, was composed of CD8⁺ T cells with

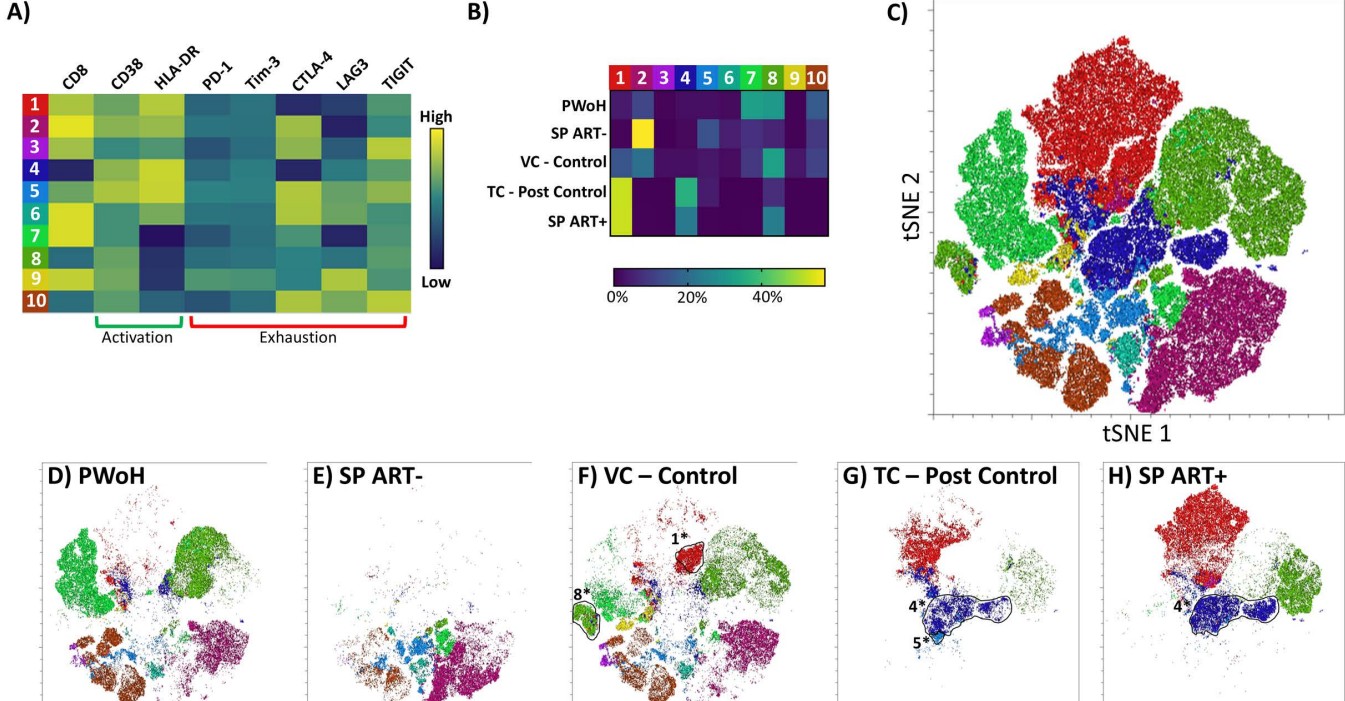

**Fig 4. Clustering and tSNE dimensionality reduction of flow cytometry profiles of CD8⁺ T cells across participant groups.** Flow cytometry was performed on PBMC from PWoH, PC and TC during successful control (VC – control), TC post control (TC – post control), SP ART-, and SP ART + . **A)** Cells from all participant groups identified as live (Zombie Yellow⁻) T cells (CD3⁺) were clustered using the FlowSOM plugin in FlowJo, labeled 1-10, color coded, and displayed as a heatmap. **B)** The relative contribution of each cluster to the overall makeup of a given participant group was plotted as a heatmap showing the percentage of cells in each participant group from a given cluster. **C)** tSNE analysis and plotting was performed on the same live T cell populations using the associated function in FlowJo v10. Final plot was colored according to the clusters identified in panel **A**. Cells from study participants corresponding to **D)** PWoH, **E)** SP ART-, **F)** VC – control **G)** TC – post control, and **H)** SP ART+ were mapped back onto the tSNE plot. Circled populations are those unique to either VC – control, TC – post control, or SP ART+ and are labelled with an identifying number and "*" that corresponds to the cluster (Fig 4A) of which they are a sub-population.

intermediate HLA-DR expression, and very low expression of PD-1, CTLA-4 and Tim-3 (Fig 5). Population 8*, another unique to VC – control, had mixed levels of HLA-DR and CD38, but expressed higher levels of exhaustion markers including CTLA-4 and LAG3 (Fig 5), similar to the flow results shown in S1 Fig. In addition, cluster 9 (Fig 4), which is unique to VC – control appears to possess a sub-population with high LAG3 expression and intermediate levels of CTLA-4 (Fig 5). TC – post control displayed profiles similar to SP ART+ characterized by high levels of activation, but also included unique populations not seen in PWoH or SP ART- (populations 4* and 5*) which were likely CD4⁺ T cells (CD3⁺, CD8⁻) with higher levels of HLA-DR and CD38, although variability was still high within the population (i.e., HLA-DR) (Fig 5). Population 4* was also present in SP ART + , but population 5*, which express Tim-3, TIGIT, and CTLA-4 but not PD-1, was absent in these SP ART + . Taken together, these data support the idea that HIV-1 controllers maintain a low level of CD8⁺ T cell exhaustion as characterized by the low abundance of PD-1⁺ and Tim-3⁺ CD8⁺ T cells, and upon loss of control and subsequent ART treatment, display activation and exhaustion phenotypes similar to SP ART + .

## Exhaustion levels remain altered over time on ART in HIV-1 transient controllers post-control

It has been observed that ART does not fully restore the polyfunctional, proliferative, or cytotoxic capabilities of dysfunctional HIV-specific CD8⁺ T cells and time from infection to ART treatment impacts the ability of ART to reduce exhaustion

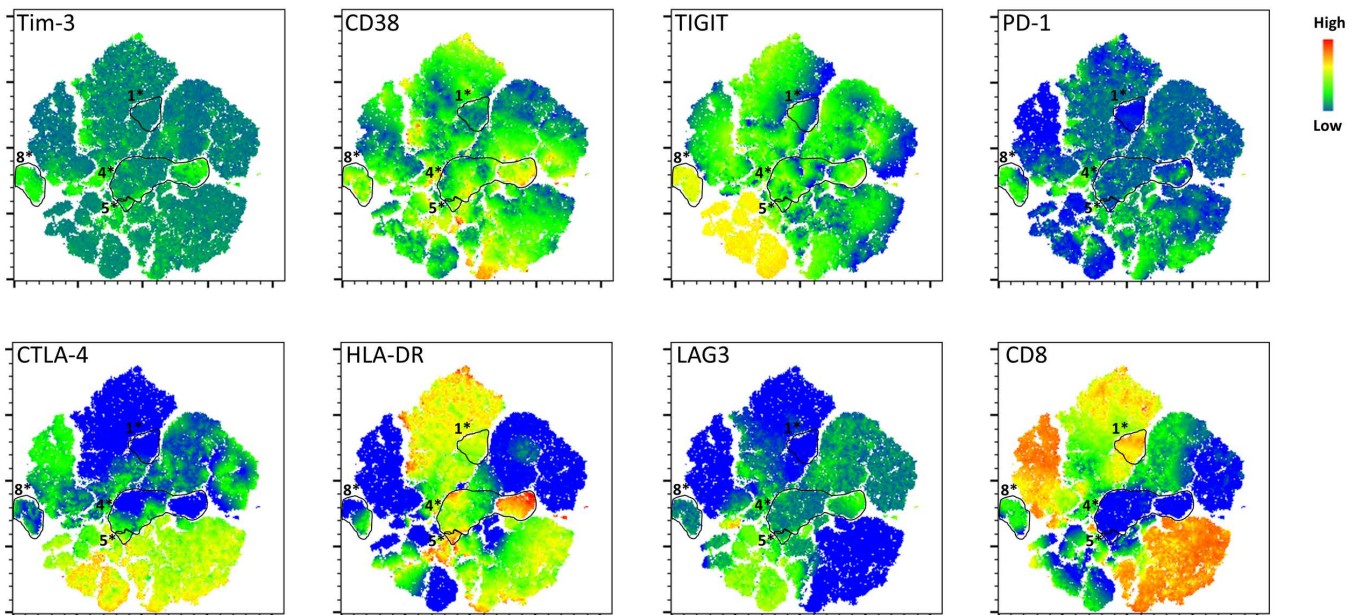

**Fig 5. Expression of individual markers by unique populations as defined by tSNE.** The intensity of staining for each individual marker was mapped back onto the tSNE plot generated in Fig 4. Heatmap indicates the relative expression levels of each marker from low (blue) to high (red). The circled and labeled populations are those that were unique to VC – control (1* and 8*), TC – post control (4* and 5*), and SP ART+ (4*) and are identified according to the cluster they were assigned to in Fig 4.

levels [64–66]. However, some studies have suggested treatment with ART is associated with reduced PD-1 expression on CD8+ T cells [50,60–62,67–69], while some suggest that Tim-3 expression persists despite treatment with ART [70–72]. One study revealed an increase in PD-1+ CD8+ T cells after HIV-1 controllers lost control, but it is unclear whether these changes persist over time after the loss of control, and how prolonged time on ART might affect exhaustion levels [20]. Therefore, we sought to determine whether the increase in Tim-3 and PD-1 expression on CD8+ T cells observed after the loss of *in vivo* viral control (Fig 3) persisted over time once these study participants were treated with ART. Once ART is initiated, the rate of viral decay varies across individuals, but typically occurs in three phases [73–75]. We analyzed exhaustion levels corresponding to three phases based on the longitudinal data available to us: < 180 days (P1), 181−365 days (P2), and 366 + days (P3) to determine the effects of ART on CD8+ T cell exhaustion over time in VC – control, TC – post control (who lost *in vivo* and *ex vivo* control), and SP ART+ (Fig 6A–6D). Significant increases in the percentage of PD-1+Tim-3- (p = 0.03) (Fig 6B) and PD-1+Tim-3+ (p = 0.007) (Fig 6D) CD8+ T cells were observed after, or just prior to one year on ART, respectively, compared to VC – control and these levels were not significantly different from SP ART+ (p = 0.08, p = 0.3 respectively) (Fig 6B and 6D), suggesting populations of exhausted T cells persist despite ART. Additionally, all TC – post control time points as well as SP ART+ displayed significantly lower percentages of non-exhausted PD-1-Tim-3- CD8+ T cells (p = <0.0001, p = <0.0001, p = <0.0001, p = 0.0007 respectively) (Fig 6A). SP ART+ also showed significantly higher levels of PD-1-Tim-3+ CD8+ T cells compared to VC – control (p = 0.04) (Fig 6C), further suggesting ART may not fully reverse CD8+ T cell exhaustion states. From these observations, it appears that levels of exhaustion do not change over time on ART within the TC – post control group.

We next aimed to determine whether the exhaustion profile observed in a subset of TC – post control who maintained *ex vivo* control [19] changed over time on ART. To evaluate this, we again grouped longitudinal exhaustion data according to the previously defined phases: <180 (P1), 181−365 days (P2), and 366 + days (P3) on ART. Interestingly, there

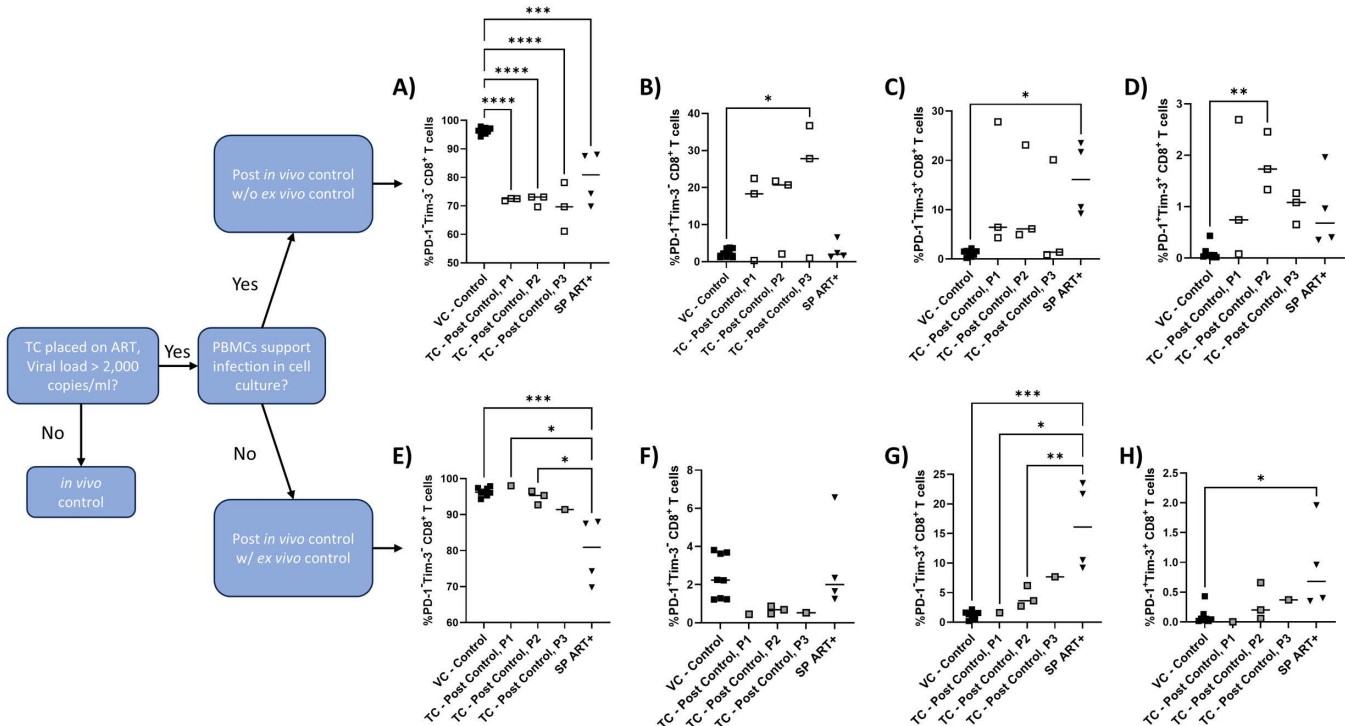

**Fig 6. CD8$^+$ T cell exhaustion levels remain altered over time on ART in HIV-1 transient controllers.** PBMCs were isolated from blood samples of PC/TC during successful control (VC – control) (n = 5-8), TC – post control < 180d ART (TC – post control, P1) (n = 1-3), TC – post control 181-365d ART (TC – post control, P2) (n = 3), TC – post control > 365d ART (TC – post control, P3) (n = 1-3), or SP ART+ (n = 4) and flow cytometry was performed gating for live (Zombie Yellow$^-$), CD8$^+$ T cells (CD3$^+$CD8$^+$). Graphed are the percentages of live CD8$^+$ T cells that are **A)** PD-1$^-$Tim-3$^-$, **B)** PD-1$^+$Tim-3$^-$, **C)** PD-1$^-$Tim-3$^+$, **D)** PD-1$^+$Tim-3$^+$, E) PD-1$^-$Tim-3$^-$, **F)** PD-1$^+$Tim-3$^-$, **G)** PD-1$^-$Tim-3$^+$, and **H)** PD-1$^+$Tim-3$^+$. **A-D)** TC – post control represent TC who lost control *in vivo* and *ex vivo*. **E-F)** TC – post control represent TC who lost control *in vivo* but maintained *ex vivo* control. Ordinary one-way ANOVA with Tukey's multiple comparisons test was performed to determine statistical significance. * p ≤ 0.05, ** p ≤ 0.01, *** p ≤ 0.0005, **** p ≤ 0.0001.

was no significant difference in the levels of PD-1$^+$Tim-3$^-$ (p = 0.7, p = 0.5, p = 0.8 respectively) (Fig 6F), PD-1$^-$Tim-3$^+$ (p => 0.9, p = 0.8, p = 0.5 respectively) (Fig 6G), nor PD-1$^+$Tim-3$^+$ (p => 0.9, p = 0.9, p => 0.9 respectively) (Fig 6H) CD8$^+$ T cells across all time points for TC – post control who maintained *ex vivo* control in comparison to VC – control, suggesting the ability to control *ex vivo* is associated with low exhaustion levels despite the loss of ability to control *in vivo*. It is also notable that exhaustion levels were generally lower in TC – post control who maintained *ex vivo* control and VC – control compared to SP ART+ as seen by lower levels of PD-1$^-$Tim-3$^+$ (p = 0.0003, p = 0.03, p = 0.01 respectively) (Fig 6G), PD-1$^+$Tim-3$^+$ (p = 0.04) (Fig 6H), and conversely, higher levels of PD-1$^-$Tim-3$^-$ (p = 0.0009, p = 0.04, p = 0.01 respectively) (Fig 6E) CD8$^+$ T cells. These results further display the association between low CD8$^+$ T cell exhaustion levels and the ability to control viral replication both *in vivo* and *ex vivo*.

We also wanted to determine the effects of exhaustion levels on CD8$^+$ T cell functionality. In order to correlate HIV-1 *ex vivo* control, clinical phenotypes, and CD8$^+$ T cell exhaustion to CD8$^+$ T cell function, we measured the production of cytokines from HIV-responsive CD4$^+$ and CD8$^+$ T cells from SP ART-, VC – control, and TC – post control. PBMCs from the indicated donor groups were unstimulated (control), stimulated with PMA/Ionomycin, or with Gag peptide pool in the presence of co-stimulation with anti-CD28/CD49d antibodies for 1h, then treated with GolgiPlug to block cellular secretory pathways, resulting in the accumulation of intracellular cytokines that can be detected by flow cytometry (S2A Fig). We observed robust general IL-2, TNF-α, and IFN-ɣ production from CD4$^+$ and CD8$^+$ T cells stimulated with PMA/Ionomycin and detectable, however, lower magnitude HIV-specific IL-2, TNF-α, and IFN-ɣ production from CD4$^+$ and CD8$^+$ T

cells stimulated with Gag peptides, indicating the presence of HIV-responsive cells (S2B–S2M Fig). We did not observe any statistically significant changes in cytokine production from either cell type when comparing responses within donor groups, however. Taken together, we observe no correlation between cytokine production from HIV-responsive cells and clinical classifications of donors based on the state of control or ART.

## Immune checkpoint blockade restores *ex vivo* control in pure T cell cultures

Since we observed a concurrence of CD8$^+$ T cell exhaustion and the loss of control in our TC cohort, we hypothesized that ICB targeting exhaustion markers could potentially restore *ex vivo* control in these participants. To test this hypothesis, and further explore whether CD8$^+$ T cell exhaustion mediates the loss of *ex vivo* viral control, activated CD8$^+$ T cells were treated with anti-PD-1 and anti-Tim-3 ICB antibodies and evaluated for their ability to suppress HIV-1 replication *ex vivo* (Fig 7A) [76,77]. Specifically, CD8$^+$ T cells and autologous CD4$^+$ T cells were isolated using magnetic bead based negative isolation, induced with PHA-P for 48 h and co-cultured at a 1:1 ratio in the presence or absence of ICB (Fig 7A). Induction of isolates with PHA-P resulted in a minimal change in viability (ratio of viability) compared to unstimulated cells (CD4$^+$− 1.187 and CD8$^+$ − 0.98; S3 Fig). Cultures were infected with HIV-1 NL4−3 and HIV-1 p24 concentrations were measured via ELISA as a surrogate marker of viral replication, from supernatants collected every 48 h post infection (Fig 7A). When anti-PD-1 (Fig 7B) or anti-Tim-3 (Fig 7C) ICB were used alone, there was no notable decrease in supernatant p24, indicating no suppression of HIV-1 viral replication.

To test the combination of ICB (cICB), anti-Tim-3 was titrated in combination with 50 ng/mL anti-PD-1 ICB (Fig 7D). Interestingly, the lowest dose of anti-Tim-3 (0.5 ng/mL) in combination with anti-PD-1 resulted in an approximate 50% suppression of viral replication compared to 50 ng/mL anti-PD-1 alone (Fig 7D, light gray triangles). Near full suppression of viral replication was seen in the 50 ng/mL anti-Tim-3 and 50 ng/mL PD-1 dosing (Fig 7D, black triangles). Remarkably, treatment with anti-Tim-3 and anti-PD-1 cICB significantly restored the ability to suppress viral replication in CD4$^+$/CD8$^+$ T cell co-cultures from TC – post control who lost the ability to suppress viral replication *ex vivo* (p = 0.003) (Fig 7E). This data directly suggests that T cell exhaustion plays a critical role in HIV-1 controllers' ability to suppress viral replication.

## Response to immune checkpoint blockade is associated with CD8$^+$ T cell exhaustion levels in PBMC cultures

CD8$^+$ T cell functionality is dependent on a complex array of signals from the surrounding microenvironment that cannot be fully recapitulated in purified T cell culture models [78–80]. To better capture the effects of these important signaling interactions, we aimed to determine whether the restored CD8$^+$ T cell functionality gained with the treatment of cICB in CD4$^+$/CD8$^+$ T cell co-cultures would also exist in the context of total PBMCs. To examine this, PHA-P stimulated PBMCs from TC – post control and an elite controller on therapy were cultured in the presence or absence of 50 ng/mL anti-PD-1 and 50 ng/mL anti-Tim-3 cICB for 24 h and then were infected with HIV-1 NL4−3. Stimulation of PBMCs resulted in a minimal change in viability (ratio of viability) compared to unstimulated cells (0.89; S3 Fig). In the absence of cICB, no participants displayed the ability to suppress viral replication *ex vivo* (Fig 7F). However, there was a dichotomous response of PBMCs to cICB across individuals. Interestingly, the ability to suppress viral replication was restored in the two TC – post control participants (VQY4910 and MPY1313) but not in the EC (EXT1011) (Fig 7F). This was unexpected since the ability to suppress viral replication was restored in this EC participant when tested in the CD4$^+$/CD8$^+$ T cell co-culture model (Fig 7D).

Since the TC – post control participants responded to cICB in both the CD4$^+$/CD8$^+$ co-culture model and in mixed PBMCs, it appears that the difference in cICB responsiveness observed is due to donor-specific attributes, and not the model system. To better understand this differential response, we directly compared CD8$^+$ T cell exhaustion levels in VQY4910 and MPY1313, the cICB responsive participants, and EXT1011, the cICB non-responsive participant, via flow cytometry. Heterogeneity was seen within cICB responsive participants, as VQY4910 had a significantly higher

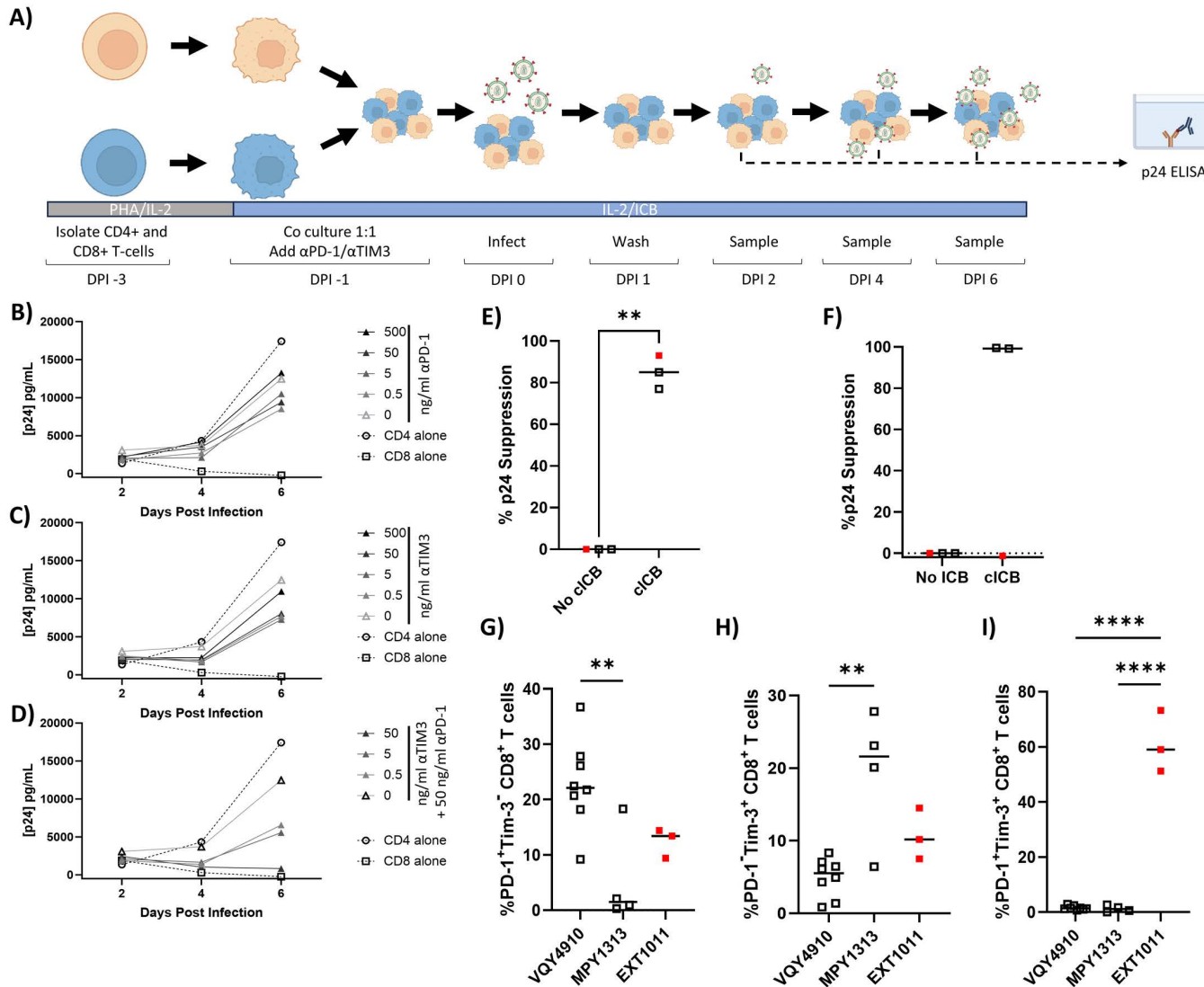

**Fig 7. Combination immune checkpoint blockade restores *ex vivo* control of viral replication in HIV-1 transient controllers – post control and CD8⁺ T cell exhaustion profiles are associated with cICB responsiveness. A)** CD4⁺ and CD8⁺ T cells were isolated from PBMC of TC – post control (n = 2) and EC (n = 1). CD4⁺ and CD8⁺ T cells were activated by PHA-P for 48 h and cultured alone or in a 1:1 co-culture and infected with HIV-1 NL4−3 after pre-treatment with ICB antibodies. Cultures were maintained in the given concentration of **B)** anti-PD-1 antibody alone, **C)** anti-Tim-3 antibody alone, or **D)** anti-Tim-3 antibody with 50 ng/mL anti-PD-1. Supernatant levels of HIV-1 p24 were determined by ELISA on days 2, 4 and 6 post infection. Ability of 50 ng/mL anti-PD-1 and 50 ng/mL anti-Tim-3 cICB to restore suppression of viral replication is shown for **E)** purified CD4⁺/CD8⁺ T cell co-cultures and **F)** total PBMCs and was determined by computing the percentage of viral suppression at peak p24 levels as compared to cICB untreated cultures. TC – post control are indicated as black symbols and the EC is indicated as a red symbol. PBMCs from blood draws performed at multiple time points following the beginning of ART in TC – post control (VQY4910 and MPY1313) and the EC (EXT1011) were examined by flow cytometry for the percentage of **G)** PD-1⁺Tim-3⁻, **H)** PD-1⁻Tim-3⁺ and **I)** PD-1⁺Tim-3⁺ CD8⁺ T cells. **E, F)** Paired two-tailed T tests were performed to determine statistical significance. **G-I)** Ordinary one-way ANOVA with Tukey's multiple comparisons test was performed to determine statistical significance. ** $p \leq 0.01$, **** $p \leq 0.0001$.

percentage of PD-1$^+$Tim-3$^-$ CD8$^+$ T cells (p = 0.007) (Fig 7G) and a significantly lower percentage of PD-1$^-$Tim-3$^+$ CD8$^+$ T cells compared to MPY1313 (p = 0.002) (Fig 7H). Assessing the more severely exhausted phenotype, we observed a significantly higher enrichment of PD-1$^+$Tim-3$^+$ CD8$^+$ T cells from EXT1011 than both MPY1313 and VQY4910 (p = <0.0001, p = <0.0001 respectively) (Fig 7I).

We next evaluated some common clinical markers of disease progression to identify any potential reasons for the observed variation in cICB responsiveness (Table 3). Interestingly, there was a large difference in the length of time on ART between the cICB-responsive and non-responsive groups with VQY4910 and MPY1313 being on ART for 206 days and 184 days, respectively, and EXT1011 being on ART for 1067 days (Table 3). It was also apparent that EXT1011 had the lowest CD8$^+$ T cell count and highest CD4:CD8 ratio (Table 3). This fact might explain the reason for responsiveness in the co-culture model, but not in mixed PBMCs as in the co-culture model, CD8$^+$ T cells are isolated and a known amount are cultured in a 1:1 ratio with CD4$^+$ T cells while in mixed PBMCs, the low CD8$^+$ T cell count and high CD4$^+$ T cell ratio would be more apparent as no pre-isolation and enrichment occurs. This would cause EXT1011 to have significantly less CD8$^+$ T cells in the PBMC mix compared to the cICB responsive donors and as our previous results have indicated, CD8$^+$ T cells are necessary for suppressing viral replication *in vitro* [19]. However, whether the extreme exhaustion phenotype, ART treatment history, CD8$^+$ T cell count, CD4:CD8 ratio, or a combination of these factors explain this participant's cICB non-responsiveness ultimately remains unclear and would require further investigation with a larger cohort.

## Discussion

Our group previously demonstrated that HIV-1 controllers who maintain the ability to suppress viral replication *in vivo* display this same ability *ex vivo* in a CD8$^+$ T cell dependent manner [19], building upon previous studies that reported similar results [20,28]. Preliminary data from our previous work suggested that this ability was tied to effective regulation of CD8$^+$ T cell activation and exhaustion. In the current study, a cohort of HIV-1 controllers who lost the ability to control viral replication presented a unique opportunity to observe longitudinal changes in T cell activation and exhaustion associated with this phenomenon. Previous studies have reported mixed results on whether CD8$^+$ T cell activation in HIV-1 controllers characterized by expression of activation markers CD38 and HLA-DR is increased or decreased [50,58,81]. Our data suggests CD8$^+$ T cell activation is low in HIV-1 controllers as CD38/HLA-DR levels were indistinguishable from those of PWoH, while levels in SP ART- were significantly higher (Fig 1). An identical pattern was observed when analyzing levels of exhaustion markers PD-1 and Tim-3 (Fig 2), which is consistent with previous reports [58,82]. Interestingly, the same pattern was not observed when analyzing levels of other exhaustion markers like CTLA-4 and LAG3, where levels in HIV-1 controllers were higher than in PWoH and SP ART-, as well as TIGIT where levels in HIV-1 controllers and SP ART- were lower than PWoH (Fig 3). These results contrast those suggesting CTLA-4 levels are unaltered in HIV-specific CD8$^+$ T cells from HIV-1 controllers [46], although since our experiments were not designed to distinguish HIV-specific versus non-specific cells, it is possible this existed in our cohort but could not be detected. One study showed that LAG3 was unaltered in gag-specific CD8$^+$ T cells from HIV-1 elite controllers [83], but since our study mostly examined bulk CD8$^+$ T cells from viremic controllers it is not surprising that our results are unique. As for TIGIT, our results are consistent with a report that TIGIT was upregulated in CD8$^+$ T cells from HIV-1 chronic progressors compared to controllers [84].

**Table 3. Clinical characteristics of cICB-responsive and non-responsive participants.**

| Classification | Participant ID | cICB responsive in co-culture? | cICB responsive in PBMC? | CD4 count (cells/mm³) | CD8 count (cells/mm³) | CD4:CD8 ratio | Days on cART |
|---|---|---|---|---|---|---|---|
| **TC** transient controllers | VQY4910 | yes | yes | 1,025 | 735 | 1.39 | 206 |
| | MPY1313 | yes | yes | 571 | 954 | 0.6 | 184 |
| **EC** elite controllers | EXT1011 | yes | no | 547 | 341 | 1.6 | 1,067 |

These data, along with the current literature, suggest that there is a relationship between the maintenance of CD8[+] T cell exhaustion and the ability to control viral replication in HIV-1 controllers [20,58,81]. To further elucidate this relationship, we monitored activation and exhaustion marker expression during periods of successful control and after the loss of control. Interestingly, we saw no change in activation marker expression after the loss of control (Fig 3), contrasting with a similar report where increased CD38 expression was observed [20]. As expected, we saw increased levels of exhaustion markers PD-1 and Tim-3 after the loss of control (Fig 3), in agreement with a similar report where increased PD-1 was observed [20]. Notably, this increase in PD-1 and Tim-3 expression after the loss of control was not reversed over time on ART (Fig 6). One study reported a slight decrease (−1.8%) in PD-1 over time on ART, but the participants analyzed were HIV-1 controllers who elected to initiate ART despite maintaining control, so aside from our study, it is still poorly understood how the loss of control and subsequent ART treatment alters exhaustion marker levels [50]. Similar observations were made through clustering analysis of our flow cytometry data, where during periods of control, HIV-1 controller T cell populations closely resembled those of PWoH (Fig 4). After the loss of control, T cell populations more closely resembled SP ART +, suggesting ART does not reverse exhaustion. The clustering and multivariate analysis also allowed us to identify and characterize populations that were unique to HIV-1 controllers (Fig 5). This analysis revealed an added layer of nuance that could not be captured through traditional flow cytometry gating. Namely, we identified multiple populations unique to HIV-1 controllers that had mixed levels of activation and exhaustion markers, with some populations displaying very low levels of exhaustion, and some displaying high levels of specific exhaustion markers but not others. Since our traditional flow cytometry gating analysis suggested low levels of exhaustion in bulk CD8[+] T cells from HIV-1 controllers, it was interesting to see that at a finer level of granularity, some populations were displaying signs of exhaustion that would not otherwise be captured. From these findings, it is tempting to speculate that these smaller populations displaying unique signs of exhaustion during periods of control may represent the early stages of loss of control, although this speculation would need to be confirmed in a larger cohort with a wider array of longitudinal timepoints to more closely track the changes in these populations. A larger cohort would also allow the continued evaluation of the effects of CD8[+] T cell exhaustion phenotypes on CD8[+] T cell functionality, especially HIV-specific cytokine production, granzyme B secretion, and cytolytic activity.

Since ART did not reverse CD8[+] T cell exhaustion (Figs 5–7), and in turn, does not restore the ability to control viral replication ex vivo [19], we aimed to determine whether immune checkpoint blockade (ICB) targeting PD-1 and Tim-3 could restore this ability. ICB is a promising therapeutic approach to restore CD8[+] T cell function that has been explored in the context of cancer as well as HIV-1 [85–87]. Intriguingly, a small sample of PLWH who received cICB treatment (targeting PD-1 and CTLA-4) for comorbid cancers saw a marked increase in CD8[+] T cell effector function [88] and significant decreases in replication competent HIV-1 [89], respectively. In addition to the potential therapeutic implications of restoring control with ICB, the ability to restore ex vivo viral control with ICB would more definitively implicate CD8[+] T cell exhaustion as the mechanism driving the loss of control in our cohort of HIV-1 transient controllers. Consistent with other studies in which blockade of PD-1 or Tim-3 alone was not sufficient to fully restore CD8[+] T cell effector function [41,42,44,70,90], monotherapy was insufficient to restore CD8[+] T cell mediated anti-HIV-1 activity in our in vitro assay (Fig 7B and 7C). However, anti-PD-1 and anti-Tim-3 combinatorial ICB fully restored viral suppression in CD4[+]/CD8[+] T cell co-cultures (Fig 7E). While it has been shown that in vitro treatment with Tim-3 ICB restored CD8[+] T cell anti-HIV-1 activity in ART naïve individuals [91], our study is the first to show that this ability can be restored in HIV-1 controllers who lost control. Also, the fact that Tim-3 ICB was not efficacious in our cohort, but was in ART-naïve individuals [91], highlights the fact that HIV-1 controllers display exhaustion and activation phenotypes unique to non-controllers. We also evaluated the effects of cICB in the context of total PBMCs and observed a varying response across participants (Fig 7E). It has been demonstrated that PD-1 and Tim-3 expression patterns (Tim-3[-]PD-1[+], Tim-3[+]PD-1[-], and Tim-3[+]PD-1[+]) are functionally distinct and identify multiple levels of T cell exhaustion [40,43,72]. This was highlighted in our observations, as the cICB-responsive donors displayed varying levels of PD-1[+]Tim-3[-] and PD-1[-]Tim-3[+] CD8[+] T cells, but showed nearly identical ability to suppress replication upon cICB treatment, while the cICB-nonresponsive donor had very high levels of PD-1[+]Tim-3[+] CD8[+] T cells, representing the most severely exhausted

phenotype, and was not able to suppress replication (Fig 7E). It is also notable that the cICB non-responsive participant was on ART for roughly 2.4 years longer than the cICB responsive participants and had the lowest CD8$^+$ T cell count (potentially skewing results in bulk PBMCs versus known CD4:CD8 ratio co-cultures) (Table 3). However, in this small cohort, it was impossible to definitively determine which of these factors contributed to the observed cICB response. Despite this, the ability of exhaustion marker-targeted cICB to restore *ex vivo* control in HIV-1 transient controllers strongly suggests CD8$^+$ T cell exhaustion is a direct contributor to the loss of control in HIV-1 transient controllers.

Some limitations to this study exist and should be considered when analyzing the results. For one, the study was limited by a relatively small cohort which hindered the ability to draw more statistically significant conclusions from our results. We were also limited to analysis of results from bulk CD8$^+$ T cells, so some granularity regarding specific CD8$^+$ T cell subsets was lost. In addition, most of our analysis was aimed at understanding changes in basal levels of activation and exhaustion, with some exploration of CD8$^+$ T cell function through understanding their ability to suppress viral replication. Future studies should aim to more fully unravel the connection between CD8$^+$ T cell exhaustion and how it alters the effector functions of unique CD8$^+$ T cell subsets in transient and persistent HIV-1 controllers.

## Conclusions

These results show that viremic HIV-1 controllers have a CD8$^+$ T cell exhaustion profile that is different and responds to treatment differently from standard progressors and elite controllers. This study suggests that disease progression has possible implications for differential responses to combinatorial ICB. In addition, our study suggests that Tim-3 and PD-1 have potential for utilization as biomarkers to screen participants with the intent of increasing the efficacy of future clinical trials investigating the use of immunotherapy as a cure strategy for otherwise healthy PLWH.

## Supporting information

**S1 Fig. HIV-1 persistent controllers display unique expression of CTLA-4, LAG3, and TIGIT.** PBMCs were isolated from blood samples of PWoH (n=2-3), VC (n=3) and SP ART- (n=9) and flow cytometry was performed gating for live (Zombie Yellow$^-$), CD8$^+$ T cells (CD3$^+$CD8$^+$). Graphed are the percentages of live CD8$^+$ T cells that are **A)** CTLA-4$^+$, **B)** LAG3$^+$, or **C)** TIGIT$^+$. Ordinary one-way ANOVA with Tukey's multiple comparisons test was performed to determine statistical significance. * p≤0.05, ** p≤0.01, *** p≤0.0005, **** p≤0.0001.
(TIF)

**S2 Fig. Measurement of HIV-specific intracellular cytokine production from CD4$^+$ and CD8$^+$ T cells from SP cART-, VCs, and TCs.** PBMCs from SP cART- (N=4), VC - control (N=5), and TC – post control (N=4) were unstimulated (control), stimulated with PMA/Ionomycin (50ng/mL and 500ng/mL, respectively) or Gag peptide pool (2μg/mL/peptide) in the presence of co-stimulatory antibodies CD28/CD49d (1μg/mL/each) for 1 h, followed by treatment with GolgiPlug for 4.5 h. Flow cytometry was performed gating for live (Zombie Yellow), CD4$^+$ T cells (CD3$^+$CD4$^+$) or CD8$^+$ T cells (CD3$^+$CD8$^+$) followed by cytokines IL-2, TNF-α, and IFN-ɣ. **A)** Representative gating of PMA/Ionomycin and Gag peptide stimulated CD4$^+$ and CD8$^+$ T cells analyzed for the expression of IL-2, TNF-α, and IFN-ɣ (Donor SRS5930, VC - control). Percentage of live CD4$^+$ T cells that are **B)** IL-2$^+$, **C)** TNF-α$^+$ (IFN-ɣ$^+$TNF-α$^+$ + IFN-ɣ$^-$TNF-α$^+$), **D)** IFN-ɣ$^+$ (IFN-ɣ$^+$TNF-α$^+$ + IFN-ɣ$^+$TNF-α$^-$), **E)** TNF-α$^+$IFN-ɣ$^-$, **F)** TNF-α$^-$IFN-ɣ$^+$, **G)** TNF-α$^+$IFN-ɣ$^+$ are quantified for control, PMA/Ionomycin, or Gag peptide treated cells. Percentage of live CD8+ T cells that are **H)** IL-2$^+$, **I)** TNF-α$^+$ (IFN-ɣ$^+$TNF-α$^+$ + IFN-ɣ$^-$TNF-α$^+$), **J)** IFN-ɣ$^+$ (IFN-ɣ$^+$TNF-α$^+$ + IFN-ɣ$^+$TNF-α$^-$), **K)** TNF-α$^+$IFN-ɣ$^-$, **L)** TNF-α$^-$IFN-ɣ$^+$, **M)** TNF-α$^+$IFN-ɣ$^+$ are quantified for control, PMA/Ionomycin, or Gag peptide treated cells. Gates were set based on the control population (CD28/CD49d treated only) such that the positive population is less than 0.1%. Data shown as means ± SD. Ordinary one-way ANOVA with Tukey's multiple comparisons test was performed to determine statistical significance within donor classifications.
(TIF)

**S3 Fig. Determination of effects of PHA stimulation on cell viability.** PBMCs (n=63), CD4+ T cell isolates (n=10), and CD8+ T cell isolates (n=55) were stimulated with PHA-P (5 µg/mL) in the presence of human rIL-2 (5 U/mL) for 48 h. Cells were subsequently washed, collected, and counted; live cells were determined by the absence of trypan blue stain. Change in viability (Δ Viability) was calculated as the ratio of the percent viability of stimulated cells (live stimulated count/ total stimulated count) to the percent viability of unstimulated (live unstimulated count/total unstimulated count) cells. Symbol colors represent individual donors and symbol shapes represent individual blood draw dates. Ordinary one-way ANOVA with Tukey's multiple comparisons test was performed to determine statistical significance. (TIF)

**S1 Data. Jones HIV controllers - supporting data values.**
(XLSX)

## Acknowledgments

We would like to thank Dr. Elias El Haddad for a critical reading of this manuscript and insightful feedback.

## Author contributions

**Conceptualization:** Amber D. Jones, Stephen Smith, ZACHARY KLASE.

**Data curation:** Zachary Capriotti, Stephen Smith.

**Formal analysis:** Amber D. Jones, Zachary Capriotti.

**Funding acquisition:** ZACHARY KLASE.

**Investigation:** Amber D. Jones, Zachary Capriotti, Erin Santos, Angel Lin.

**Methodology:** Amber D. Jones, Zachary Capriotti, ZACHARY KLASE.

**Project administration:** Rachel Van Duyne, ZACHARY KLASE.

**Resources:** Erin Santos, Stephen Smith.

**Supervision:** Rachel Van Duyne, ZACHARY KLASE.

**Writing – original draft:** Amber D. Jones, Zachary Capriotti.

**Writing – review & editing:** Angel Lin, Rachel Van Duyne, Stephen Smith, ZACHARY KLASE.

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
