## [Decision Letter · Decision Letter 0]

17 Mar 2025

PONE-D-25-04334HIV-1 controllers possess a unique CD8+ T-cell activation phenotype and loss of control is associated with increased expression of exhaustion markersPLOS ONE

Dear Dr. KLASE,

Thank you for submitting your manuscript to PLOS ONE. After careful consideration, we feel that it has merit but does not fully meet PLOS ONE’s publication criteria as it currently stands. Therefore, we invite you to submit a revised version of the manuscript that addresses the points raised during the review process.

 Kindly address all the comments raised, but pay particular attention to the comments in section 1.

We look forward to receiving your revised manuscript.

Kind regards,

Owen Ngalamika

Academic Editor

PLOS ONE

Journal Requirements:

2. In the online submission form, you indicated that values used to perform analyses and to generate graphs available from the authors upon reasonable request. Flow cytometry data files will be available through the International Society for the Advancement of Cytometry’s Flow Repository.

Reviewers' comments:

Reviewer's Responses to Questions

**Comments to the Author**

1. Is the manuscript technically sound, and do the data support the conclusions?

Reviewer #1: Partly

2. Has the statistical analysis been performed appropriately and rigorously? 

Reviewer #1: Yes

3. Have the authors made all data underlying the findings in their manuscript fully available?

Reviewer #1: Yes

4. Is the manuscript presented in an intelligible fashion and written in standard English?

Reviewer #1: No

5. Review Comments to the Author

Reviewer #1: Jones and colleagues set to address specific exhaustion marker profiles of CD8+ T cells in people with HIV. It is a nice confirmation of work that has been part of the prevailing model in CD8+ T cell function. Overall, it is a nice story, and there is a great deal of additional potential. However, there are several broad general challenges that need to be addressed (discussed first) and some initial quick points (derived early during review).

Section 1:

There is great plasticity in CD8+ T cells (central, effector, etc) but here, the granularity is lost since bulk CD8s are examined. So while exhaustion markers are used, functionality of the CD8s is the other half that seems to be missing. The study does well in providing an overall sense of basal levels but there is much more that could and probably should be explored here. It would have been nice to add HIV peptide stimulation (with controls) to further tease out how exhaustion and cytolytic function are entwined. Given the limitation in the number of participants, it is a worry on the Power to know if the differences between "high" vs. "low" levels of plasma viremia can be parsed out--clarification on whether the "low" are actually viremic controllers is needed--with the inclusion of one Elite controller is nice to have but would be nice if it were at least three. One of the biggest challenges came from the writing. The methods need to be fleshed out more. The introduction had "pacing" issues. If possible for the results, either combine sections to take more time providing results. State what p-values might be instead of just relying on the legends. There were several cases where not all of the figure panels were discussed either. Data visualization (minus the tables), was very clear (font in Fig 7 could be increased).

Section 2:

• Lines 41-57: It is commendable that the authors are trying to compact HIV clinical progression with many factors that are at play. It should be tightened and explore the space. For instance, “HIV-1 without therapy”, those familiar will know this as ART, but it would benefit to say “HIV-1 without antiretroviral therapy (ART)”. Please also be conscious to use People First language, “patient” is no longer accepted, but study participant or people/persons with HIV (PWH)—Please also see Methods as PWH are not “subjects”. The second paragraph suffers from much of the same issues as the first. It is clear the authors are being thoughtful, but this Reviewer would strongly suggest making more succinct.

• Lines 94-96: The authors do a nice job discussing Tim-3 and PD-1 but then just list other markers. Perhaps in condensing the first two paragraphs, this paragraph can be expanded.

• Line 130: Since MHC-II HLA alleles are known, suggest including in Table 1.

• Line 136: Are the LVL donors viremic controllers? Based on the cut-offs it appears so but not sure if that is your aim.

• Line 143: The authors have copies/ml and copies/mL. It is not clear why for one donor “ND” was used for CD8 count but Persons without HIV “Pw/oH” just have a dash. Some numbers have commas others do not. Cell counts need the mm-cubed to actually be a superscript. For ART regimens, it is better to spell out the non-brand name (Truvada=TDF/FTC=tenofovir/emtricitabine).

• Line 151: Out of curiosity, what was cell viability for a 48hr PHA stimulation? Typically 24hrs is more than enough before it starts to become toxic.

• Lines 162-163: More information is needed for virus collection. How was the virus “filtered”. Frozen stocks at what temperature?

• Line 170: 17ng/ml of NL4-3 is oddly very specific. How was this amount of virus determined for the infection? It would be nice to also know what the MOI was considered.

6. PLOS authors have the option to publish the peer review history of their article (what does this mean? ). If published, this will include your full peer review and any attached files.

**Do you want your identity to be public for this peer review?** For information about this choice, including consent withdrawal, please see our Privacy Policy .

Reviewer #1: No

---

## [Author Response · Author response to Decision Letter 1]

30 May 2025

Response to reviewers has been uploaded as an attachment as instructed. We thank the reviewers for their time and efforts in improving our manuscript.

---

## [Decision Letter · Decision Letter 1]

19 Jun 2025

PONE-D-25-04334R1HIV-1 controllers possess a unique CD8+ T-cell activation phenotype and loss of control is associated with increased expression of exhaustion markersPLOS ONE

Dear Dr. KLASE,

Thank you for submitting your manuscript to PLOS ONE. After careful consideration, we feel that it has merit but does not fully meet PLOS ONE’s publication criteria as it currently stands. Therefore, we invite you to submit a revised version of the manuscript that addresses the points raised during the review process.

Kindly address the minor suggestions by the reviewer on reporting of p values. Also, please ensure consistency in units of measurements (e.g. either ml or mL).

We look forward to receiving your revised manuscript.

Kind regards,

Owen Ngalamika

Academic Editor

PLOS ONE

Journal Requirements:

Reviewers' comments:

Reviewer's Responses to Questions

**Comments to the Author**

1. If the authors have adequately addressed your comments raised in a previous round of review and you feel that this manuscript is now acceptable for publication, you may indicate that here to bypass the “Comments to the Author” section, enter your conflict of interest statement in the “Confidential to Editor” section, and submit your "Accept" recommendation.

Reviewer #1: All comments have been addressed

2. Is the manuscript technically sound, and do the data support the conclusions?

Reviewer #1: Yes

3. Has the statistical analysis been performed appropriately and rigorously? 

Reviewer #1: Yes

4. Have the authors made all data underlying the findings in their manuscript fully available?

Reviewer #1: Yes

5. Is the manuscript presented in an intelligible fashion and written in standard English?

Reviewer #1: Yes

6. Review Comments to the Author

Reviewer #1: The revised manuscript by Jones and Capriotti et al. is very much improved from the original draft. It is greatly appreciated the time the authors took to address comments and rework many aspects. It is a much stronger and compelling body of work. The only minor suggestion is with p-values, sometimes the number of sig. figs is too much (e.g., p=0.0402 could just be 0.04 but p=0.0003 is 0.0003). They are not inherently wrong, but I am not sure the sample size is large enough to go to that many sig. figs.

7. PLOS authors have the option to publish the peer review history of their article (what does this mean? ). If published, this will include your full peer review and any attached files.

**Do you want your identity to be public for this peer review?** For information about this choice, including consent withdrawal, please see our Privacy Policy .

Reviewer #1: No

---

## [Author Response · Author response to Decision Letter 2]

3 Jul 2025

We have adjusted p-values to one significant figure and changed all instances of 'ml' to 'mL' for consistency. We thank the reviewers for their time and efforts in improving our manuscript.

---

## [Editor Report · Decision Letter 2]

6 Jul 2025

HIV-1 controllers possess a unique CD8+ T-cell activation phenotype and loss of control is associated with increased expression of exhaustion markers

PONE-D-25-04334R2

Dear Dr. KLASE,

We’re pleased to inform you that your manuscript has been judged scientifically suitable for publication and will be formally accepted for publication once it meets all outstanding technical requirements.

Kind regards,

Owen Ngalamika

Academic Editor

PLOS ONE
---

## [Editor Report · Acceptance letter]

PONE-D-25-04334R2

PLOS ONE

Dear Dr. KLASE,

I'm pleased to inform you that your manuscript has been deemed suitable for publication in PLOS ONE. Congratulations! Your manuscript is now being handed over to our production team.

Kind regards,

on behalf of

Dr. Owen Ngalamika

Academic Editor

PLOS ONE